# Stressors, Psychological States, and Relationship Quality among East Malaysian Adults with Partners Amid the COVID-19 Lockdown

**DOI:** 10.3390/ijerph191811258

**Published:** 2022-09-07

**Authors:** Jasmine Adela Mutang, Bee Seok Chua, Kai Yee Hon, Ching Sin Siau, Walton Wider, Rosnah Ismail

**Affiliations:** 1Faculty of Psychology and Education, Universiti Malaysia Sabah, Kota Kinabalu 88400, Sabah, Malaysia; 2Centre for Community Health Studies, Universiti Kebangsaan Malaysia, Bangi 43600, Selangor, Malaysia; 3Faculty of Business and Communications, INTI International University, Nilai 71800, Negeri Sembilan, Malaysia; 4Faculty of Allied Health Sciences, University of Cyberjaya, Persiaran Bestari, Cyber 11, Cyberjaya 63000, Selangor, Malaysia

**Keywords:** adults with partners, stressors, mental health, relationship quality, COVID-19

## Abstract

The COVID-19 pandemic has had a huge impact not only on healthcare systems and economic threats but also on relationships. Due to the massive measures of the Movement Control Order, such as social distancing, strictly limited physical activities, and making adjustments to working from home—merged with the pandemic’s fear—romantic partners may face challenges that affect the quality of their relationships. The purpose of this study was to (i) identify stressors experienced by participants during the COVID-19 lockdown, (ii) evaluate participants’psychological well-being before and during the lockdown, and (iii) see if participants’ perceived relationship quality differed before and during the lockdown. An online self-report questionnaire was used to recruit 334 adults (58.1 percent female, 41.9 percent male) with partners (married = 304, engaged = 12, and in committed relationships = 17). Participants were asked about stressors they encountered as a result of COVID-19 using an open-ended question. The Depression Anxiety Stress Scales (DASS-21) were used to assess mental health, and the Perceived Relationship Quality Component (PRQC) Inventory was used to assess relationship quality. According to the findings, the top three common stressors reported by participants were financial problems, restricted movement, and fear of COVID-19 infection. Participants reported significantly higher levels of stress, anxiety, and depression during the lockdown than before. The findings also revealed that participants’ relationship quality improved during the lockdown compared to before the lockdown. The current study contributes by providing information on the impact of the COVID-19 pandemic on mental health and the quality of couples’ relationships during the pandemic.

## 1. Introduction

During the period of the COVID-19 pandemic, people were exposed to a number of traumatic events and circumstances. These included social isolation, financial difficulties, witnessing death or suffering, a mutation of the COVID-19 virus, a healthcare system on the verge of collapse, an increase in new daily cases that exceeded the maximum capacity of hospitals, sensationalism by the mass media, and a number of other factors. Stressful situations and circumstances touched nearly every person, either directly or indirectly, in some way. As a consequence of this, the effect of the pandemic on the mental health outcomes of the general population has been investigated rather frequently.

In a study that was carried out on 1468 members of the general population in the United States in April 2020, [1] revealed that 13.6% of people in the United States had signs of substantial psychological distress, which is an increase from 2018. The authors of [2] conducted a study on 9565 members of the general population from 78 different countries during the pandemic lockdown that lasted from April to June in the year 2020. According to their findings, the majority of people throughout the pandemic period experienced moderate levels of stress, while 11% reported experiencing the greatest levels of stress. In addition to reporting high levels of depression and boredom, respondents also indicated that the activities they participated in did not provide any form of reinforcement for them, and nearly half of the respondents reported wasting a significant amount of their time. During the early stages of the COVID-19 pandemic (March–July 2020), Randall et al. conducted a study that examined the impact of the pandemic not only on psychological consequences but also on the relationship quality among spouses [3]; the study involved 14,020 people from 27 different countries. They reported that participants experienced increased psychological distress after COVID-19 country-level restrictions were implemented compared to before, and reports of psychological distress were associated with lower relationship quality. This was the case in the majority of countries, but not all of them.

The purpose of the study conducted by [4] was to investigate the perceived levels of psychological distress and relationship quality among 124 married couples in Malaysia both before and during the lockdown, as well as factors that were predictive of the participants’ levels of well-being. They came to the conclusion that during the COVID-19 epidemic, spouses displayed a greater degree of psychological anguish. Their discovery was consistent with the findings of [5] studies that were carried out on the general populations of China, Italy, Iran, Spain, Turkey, Nepal, the United States of America, and Denmark [6]. Günther-Bel et al. discovered that married people experienced higher levels of psychological suffering during the lockdown in Spain [5], which was particularly true for families with children. Furthermore, according to the findings of [4], positive predictors of well-being in a relationship include feelings of contentment and trust, as well as sexual activity. It is interesting that they found that during the pandemic lockdown, the couples they studied seemed to acquire more trust in one another. Another study by Chua et al. was carried out on 543 members of the general public in Malaysia between June 2020 and October 2020 [7]. The purpose of this study was to investigate the effects of COVID-19 on lifestyle and how it affects the well-being of Malaysian adults. According to their [7] findings, the COVID-19 pandemic had an impact on various facets of daily life, including the fact that most people reported increased levels of internet and social media usage, working from home, performing household chores, resting and relaxing, but participated in sports less frequently than they had before the pandemic.

In general, research has shown that adults have had significant levels of anxiety and conditions connected to COVID-19 [8,9]. The majority of research has been cross-sectional, and many studies have focused on subgroups that are more likely to be affected by the disease (for example, healthcare and vital employees [10]). Therefore, the purpose of this study is to extend the findings of [7] by using a larger sample size to investigate the stressors that were experienced by adults with partners, their psychological wellbeing during the lockdown, and to find out how participants perceived the quality of their relationship with their partner prior to and during the pandemic lockdown, with a particular emphasis on East Malaysian adults. Before the nationwide Movement Control Order in Malaysia that was issued as a result of the COVID-19 outbreak, the states of Sabah and Sarawak in Eastern Malaysia had instituted their own travel restrictions. They established a mandatory 14-day home quarantine for state nationals who had returned from China and prevented foreigners from entering mainland China or having a history of travel there. These were issued by the Sabah State Secretary, appointed in 2020 [11], and the Sarawak State Disaster Management Committee, appointed in 2020 [12]. It is possible that the consequences of the COVID-19 pandemic on the populations of these two states would be different from those experienced by Western Malaysians because of the disparities in the preventative measures taken by the two states.

## 2. Method

### 2.1. Participants

A total of 334 adults (58.1% female, 41.9% male) with partners (married, engaged, or in a committed relationship), aged between 21 and 68 years (*M* = 40.0, *SD =* 10.8) were involved in the study. Most of the participants were married (91.3%), 5.1% were in a committed relationship, and the remaining 3.6% were engaged. The average years of staying together for married couples was 14.3 years (*SD* = 11.0); for being engaged (*M* = 2.7, *SD* = 7.1); for committed relationships (*M* = 10.2, *SD* = 9.7). The average number of children was 2.2 people (*SD* = 1.7). Table 1 presents the characteristics of the participants of the study.

#### Measures

*Background Information.* The first part of the questionnaire included a section on background information (gender, relationship status, years of partnership, and the number of children).

*Stressors related to COVID-19*. Participants were asked to respond to an open-ended question on what stressors they experienced as a result of COVID-19 during the lockdown. “What stressors are you experiencing due to the COVID-19 during the national lockdown?”

*Mental Health.* The Depression Anxiety Stress Scale-21 (DASS-21) was developed by [13] and consists of 21 items. Each sub-scale is measured by seven items: depression (7 items), anxiety (7 items), and stress (7 items). Participants were asked to rate the severity of each symptom over the previous week using a 4-point frequency scale (0 = never, 1 = occasionally, 2 = frequently, and 3 = almost usually). Higher scores indicate more frequent symptomatology.

*The Perceived Relationship Quality Component (PRQC) Inventory.* To assess relationship quality, the PRQC Inventory [14] with 18 items was employed. On each statement, participants were asked to rank their present romantic partner and relationship. Each statement is answered on a 7-point Likert-type scale (ranging from 1 = not at all, to 7 = extremely). The PRQC consists of six components: (i) relationship satisfaction, (ii) commitment, (iii) passion, (iv) trust, (v) love, and (vi) intimacy. 

### 2.2. Data Collection Procedure

This study aimed to (i) identify the stressors that participants experienced during the COVID-19 lockdown, (ii) examine the participants’ psychological wellbeing before and during the lockdown, and (iii) determine whether or not the participants’ perceived relationship quality was different before and during the lockdown. The data collection began in September 2020 and lasted through December 2020 during the lockdown. The participants who met the inclusion criteria were chosen using a convenience sampling technique to complete the self-report survey, administered via the internet (e.g., e-mail and WhatsApp). The objectives of the study were explained to participants and they were given the assurance that the data collected would only be utilised for research purposes. Participants were made aware that taking part in the study was entirely up to them and that they were free to revoke their permission at any moment. 

### 2.3. Data Analysis

The responses to the open-ended question that asked participants to describe the sources of stress they experienced during the COVID-19 lockdown were analysed using a multilevel coding method [15]. The question asked participants, “What stressors are you experiencing due to the COVID-19 during the national lockdown?” We retrieved the pertinent codes from the responses provided by participants and then categorised these codes based on the similarities or links between the characteristics and dimensions that they had in common. Then, in order to allow the previously identified subcategories to emerge and take shape, we carried out an ongoing comparative and progressive evaluation of the examples. Meanwhile, the quantitative data was analysed using the IBM Statistical Package for the Social Sciences (SPSS) Version 28 software (Armonk, NY, USA). Statistical significance was defined as *p* < 0.05. Descriptive data were described using frequency, means, percentages, and standard deviations. A paired sample *t*-test was used to compare participants’ psychological states and the relationship quality before and after the COVID-19 lockdown.

## 3. Results

### 3.1. Participants’ Main Stressors Related to COVID-19

During the lockdown, there were a total of 306 participants, and 91.6 percent of them responded to the open-ended question that asked about the stressors they faced linked to COVID-19. Because each open-ended response could be associated with many different stressors, the total number of replies on a single subject as a stressor was used to calculate the percentages rather than the overall number of respondents. Twelve different themes surfaced during the multilevel coding analysis. The top five most prevalent sources of stress were finance-related problems (21.3%), restricted movement (20.68%), fear of COVID-19 transmission to self and others (14.2%), work-related challenges (12.04%), and social restrictions (7.10%). The details of the common sources of anxiety that participants encountered during the lockdown are presented in Table 2. 

### 3.2. Psychological States before and during the COVID-19 Lockdown

The paired samples *t*-tests were used to analyse the perceived psychological states of participants in Eastern Malaysia’s COVID-19 lockdown both before and during. During the lockdown, participants’ levels of depression (t = −8.93, *p* < 0.05), anxiety (t = −10.17, *p* < 0.05), and stress (t = −9.95, *p* < 0.05) were shown to be significantly different from those measured before the lockdown. When compared to their levels before the lockdown, participants in the COVID-19 study had considerably higher levels of sadness, anxiety, and stress during the lockdown (refer to Table 3).

#### Relationship Quality before and during the Lockdown

We used paired sample *t*-tests to examine how participants felt about the quality of their relationships both before and after the lockdown. The findings showed that the quality of the relationship was significantly different prior (*M* = 111.25, *SD* = 13.91) and subsequent to (*M* = 112.19, *SD* = 13.63; *t* (333) = 2.61, *p* = 0.009) the lockdown. During the lockdown, they had an overall improvement in the quality of their relationships, particularly in the areas of commitment, trust, passion, love, and sex. The only areas where they did not experience improvements were satisfaction and intimacy. In Table 4, the quality of the relationship is displayed both before and after the lockdown.

## 4. Discussion

According to the findings, the five most common sources of stress for adults with partners were as follows: (1) financial problems, (2) stress from restricted movement, (3) fear of COVID-19 transmission, (4) work-related challenges, and (5) social restrictions. There is little doubt that the pandemic affected the majority of breadwinners in the house since the employment impact was drastically decreased in most industries after the Movement Control Order (MCO) during the months of September and October in the year 2020 [16]. For one participant, a member of their family found themselves suddenly unemployed as a result of the epidemic, which led to the family’s financial instability and inability to provide for all members. Because the cost of living in Sabah is significantly greater than in the majority of other states in Malaysia, the family had a difficult time making ends meet. In addition, the restriction of movement had an effect on the quality of relationships within families. The participants reported that they were unable to see their children, who lived a great distance away from them, and that they were unable to engage in the forms of recreation they were accustomed to before the pandemic; both of these factors contributed to a decrease in the quality of family relationships. It was made abundantly evident that the way of life for East Malaysians has been altered, and they now favour activities that take place outside and prefer to spend time with family members, particularly so on the weekends due to the abundance of nature reserves that can be found in Sabah. Their social life was indirectly impacted in the sense that the traditional beliefs and social customs that are practised in Sabah that help to shape the tolerant relationships that exist between people of many ethnicities and religions ceased [17]. Intimate family relationships, such as marriage, are the basis of tolerance among Eastern Malaysia’s various ethnic groups and they also help to strengthen the social cycle. The unexpected lifestyle changes triggered the stressors of respondents significantly. Like in many other countries, residents were fearful of COVID-19 transmission, even though a past study in Malaysia found that respondents demonstrated adequate knowledge of the spread of the virus [18], especially women. Information on how to curb COVID-19 was well established by the government, and it has become the norm to wash hands frequently, sanitize, and keep good self-hygiene. Regardless of the awareness, they still had the possibility to get infected, as respondents reported they did not know how they got infected as they stayed at home and kept clean most of the time. It is believed that the individual’s immune system is playing a vital role in protecting the human body from COVID-19 and how it helps to fight the virus [19].

In other areas, the difficulties associated with work were also reported by participants. There has been a significant shift toward computer-based working styles, and the majority of tasks have been digitised. For example, many employers now give their employees the option of working from home or the office, online meetings are commonplace, and all necessary documentation can be accessed via the internet. Those employees who were unprepared for the new technology method were subjected to a significant increase in the amount of stress they experienced on the job. The authors of [20] found that a total of 67% of Malaysians had experienced stress due to changes in their work and organisation, and 63% claimed to have problems with work-life balance. To put it another way, those who were working from home needed to work more hours than usual; instead of eight hours of working a day, fifteen hours a day was reported as the average amount of time spent working at home. In addition, parents who were employed outside the home had an additional responsibility to monitor their children’s participation in remote learning activities when at home. Where the parents performed multiple tasks at once at home, this resulted in an increased amount of effort.

In comparison to before the lockdown, participants in the COVID-19 study reported considerably higher levels of despair, anxiety, and stress when they were in lockdown. This is consistent with the findings of studies carried out by [21,22,23]. A higher degree of depression, anxiety, and stress could have been triggered by the Sabah election from 26 September 2020 to 12 October 2020, which recorded 70 percent of instances testified to become the third COVID-19 wave in Malaysia. This election took place from 26 September through 12 October 2020 [24]. Even though the virus spread quickly to other states in Malaysia over the course of the following few weeks, the residents of Sabahan (located in Eastern Malaysia) reported considerably higher levels of sadness, anxiety, and stress than their counterparts in West Malaysia. The issue of mental health is, without a shadow of a doubt, the top issue that needs to be addressed during the pandemic, as it might affect people of any age range. The present study contributes to the evidence of the previous study from [4] in that, having an even larger sample size of respondents, East Malaysian couples demonstrated similar results in their psychological well-being during the lockdown.

Moreover, the findings of the current study add new evidence to those found in earlier work by [4]. The most significant findings from this study were that participants experienced a better relationship quality during the lockdown in comparison to before the lockdown, specifically in their commitment, trust, passion, love, and sex components. The only exceptions were in the satisfaction and intimacy components. Relationship satisfaction, love, intimacy, and passion were significantly lower post-lockdown when compared to the pre-lockdown period, which means that both cohabiting and married couples were spending quality time together during the lockdown. Activities such as watching movies, doing household chores, and cooking helped to boost the love and trust between each other. In line with the study from [25], relationship satisfaction, love, intimacy, and passion were significantly lower post-lockdown compared to pre-lockdown. However, a study by [26] that included participants from Spain, Austria, Poland, and the Czech Republic, found that machine learning models did not accurately predict the quality of relationships, and that age was the most important predictor; younger ages predicted higher quality relationships. On the basis of this argument, we are able to get to the conclusion that "collectivism" is still alive and well in Eastern cultures, particularly among East Malaysians in the current research. In any circumstance, couples do require a calm atmosphere in order to keep the quality of their relationship intact. In the Western study by [27], the female participants indicated that they did not experience any differences in their libido; however, some of the female participants did report a decrease in pleasure, satisfaction, desire, and arousal during sexual activity. These findings indicate that, although Eastern nations are classified as "conservative" in terms of sexual behaviour, the married couples that participated in the current study reported higher levels of sexual satisfaction during the lockdown. Nevertheless, qualitative research is recommended for the upcoming study in order to investigate the degree to which the quality of relationships has improved.

### Limitations and Recommendations

To compare the participants’ situation before and after the lockdown, we used recall questions. While recall questions are commonly used in surveys [28] and studies on couples (e.g., [29,30]), they are prone to bias as the accuracy of respondents’ memories cannot be controlled for [31]; however, this bias might still be limited. It is becoming clear that the ongoing COVID-19 pandemic and its many “lockdowns” have adversely impacted the mental health of the general public. It has been suggested that our ability to recall autobiographical events is influenced by our prevailing affective state (e.g., mood-congruence recall bias) [32]. In this context, participants experiencing negative affect valence at the time of the survey may recall their stressors, psychological states, and relationship quality prior to a COVID-19 ‘lockdown’ with greater pessimism than those experiencing a more positive affective state.

Because we did not ask participants what stressors most influenced their relationship with their partner during the COVID-19 lockdown, future research could expand our study by including such an open-ended question to gain a more complete, balanced, and in-depth understanding of relationship quality among adults with partners.

## 5. Conclusions

In the current study, conducted during COVID-19, East Malaysian adults with partners revealed five stressors. Of these, financial problems were indicated as being the primary issue. According to the findings of this research, the pandemic is having a significant impact on mental health as well as the relationship dynamics of people. On the other hand, the quality of their connection revealed an improvement in the relationship during the lockdown. These significant findings assist to highlight that spending time together and enjoying leisure activities with one’s partner is essential, regardless of the circumstances. This was the case regardless of whether or not they had children. In addition, the findings indicate that "togetherness" is the most important factor in ensuring that adults with partners in Eastern Malaysia are able to keep their relationships of high quality. Future qualitative research is required if we are going to have a greater understanding of how to keep the quality of the connection between couples, and the results of this research could eventually become the guidelines for couples who are having issues.

## Figures and Tables

**Table 1 ijerph-19-11258-t001:** Characteristics of participants.

Variables	Frequency (*n* = 334)	Percentage	Mean	Standard Deviation
Gender				
Men	140	41.9		
Women	194	58.1		
Relationship Status				
Married	304	91.1		
Engaged	12	3.5		
Committed Relationship	18	5.4		
Age			40.0	10.8
Years of partnership				
Married			14.3	11.0
Engaged			2.7	7.2
Committed Relationship			10.2	9.7
Number of children			2.2	1.7

**Table 2 ijerph-19-11258-t002:** Reported main stressors encountered during the lockdown due to COVID-19, “What stressors are you experiencing due to the COVID-19 during the national lockdown?”.

No.	Main Stressors	*n*	(%)	Example Quotations
1.	Finance-Related Problems (e.g., job loss/pay cut/business shut down/retrenched)	69	21.30	*“Salary being cut down due to sales of room reduce (Hotel xxx)”* *“I was terminated”* *“I was really stress because I had to closedown my business”*
2.	Restricted Movement/Confinement	67	20.68	*“I’m stressed because I have to be in the house for a long time”* *“As an engineer working at a construction site, stop work order has caused a lot of delay in our work progress. Not able to exercise or go to the gym. Not able to carry out our plan for house renovation”* *“Due to the risk of contracting COVID-19, I was unable to meet with my child who lives far away”*
3.	Fear of COVID-19 transmission to self and others	46	14.20	*“Fear of the possibility of contracting COVID-19 because of instances in the neighbourhood is* on the rise” *“Fear of children and families being infected COVID-19”*
4.	Work-Related Challenges (e.g., adapting to working from home/disruptions/internet problems)	39	12.04	*“Work is difficult challenging in terms of communication because I have many overseas clients”* *“Need to work from home and have internet problems”* *“Had to work to the office and had to leave the children without guardians”*
5.	Social Restrictions	23	7.10	*“No longer allowed to engage in sports and gatherings”* *“Feeling stressed about not being able to carry out leisure activities outside the house as usual, hanging out with friends and having to always put on face mask all the time when going out”* *“Can’t go out and buy things freely, because always trying avoid the crowd”*
6.	Psychological Health Issues (e.g., anxiety/depression/emotional disturbances)	19	5.86	*“Experiencing restlessness, fear and worry”* *“I have just given birth and my mental health has been severely disrupted due to financial problems which have just started to stabilize, but are now deteriorating again due to the increasing number of cases”* *“The personal stress where I always struggle with feeling tired and uncomfortable due to the lack of frequency to go out of the house”*
7.	Fear of uncertainty	11	3.40	“*Worry and anxious about the absence of vaccine and the increase of COVID-19 cases”*
8.	Stress adapting to new norms	15	4.63	*“Distressed about the new norm of not being able to do regular business at the farmer’s market”*
9.	Burnout/Workload	13	4.01	*“Heavy workload but difficult to work from home”*
10.	Other concerns (e.g., fear of losing job/disrupted plans/helplessness/feeling overwhelmed)	12	3.70	*“Fear of losing my job if this situation still persists”*
11.	Other health issues (e.g., treatments postponed/sleep problems/health affected)	9	2.78	*“Concern about nm monthly therapy and medicine has been postponed”*

Note: Percentages shown are calculated using the total number of responses rather than the total number of respondents.

**Table 3 ijerph-19-11258-t003:** Pair samples *t*-tests of participants’ psychological states before and during COVID-19 lockdown.

Variables	N	Mean	Std. Deviation	t	Sig.
Depression During Lockdown	334	2.14	3.11	−8.93	<0.001
Depression Before Lockdown	334	3.30	3.76		
Anxiety During Lockdown	334	2.45	3.02	−10.17	<0.001
Anxiety Before Lockdown	334	3.75	3.77		
Stress During Lockdown	334	3.57	3.77	−9.95	<0.001
Stress Before Lockdown	334	4.98	4.40		

**Table 4 ijerph-19-11258-t004:** Relationship Quality Before and During the COVID-19 Lockdown.

Items	*n*	* M *	* SD *	* t *	* p *
Overall relationship quality before lockdown	334	111.25	13.91	−2.61	0.009
Overall relationship quality during lockdown	334	112.16	13.63		
Satisfaction before lockdown	334	18.53	2.88	1.08	0.282
Satisfaction during lockdown	334	18.42	2.93		
Commitment before lockdown	334	18.85	2.57	−3.55	<0.001
Commitment during lockdown	334	19.11	2.45		
Intimacy before lockdown	334	18.90	2.69	−1.91	0.057
Intimacy during lockdown	334	19.05	2.58		
Trust before lockdown	334	18.73	2.56	−2.64	0.009
Trust during lockdown	334	18.91	2.57		
Passion before lockdown	334	16.95	3.39	−3.06	0.002
Passion during lockdown	334	17.24	3.41		
Love before lockdown	334	19.28	2.53	−2.73	0.007
Love during lockdown	334	19.46	2.45		
Sex before lockdown	334	37.23	4.80	−2.61	0.009
Sex during lockdown	334	37.59	4.70		

## Data Availability

The data presented in this study are available on request from the corresponding author.

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
