# Peer review of "Stressors, Psychological States, and Relationship Quality among East Malaysian Adults with Partners Amid the COVID-19 Lockdown"

_ijerph, 2022, doi:10.3390/ijerph191811258_

Round 1

Reviewer 1 Report

The study is a cross-sectional survey. The topic about relationship quality might be of interest of readers. The manuscript has focused on stressors, psychological well-being and relationship. It might be a topic that fits the journal. I have these issues for your consideration. I will suggest a revision and resubmission.

[1]You seemingly suggest an intervention is needed but your findings suggest the relationship has been improved during lock down. So maybe you do not need an intervention for relationship?

[2]It is better to have citations for this sentence: "Their discovery was consistent with the findings of other studies that were carried out on 69 the general populations of China, Italy, Iran, Spain, Turkey, Nepal, and the United States 70 of America and Denmark."

[3] It is not easy to find confirmation whether you measured both before and during lockdown status for all your measurements (psychological well-being and relationship) or relationship quality only. Your wording suggested it is relationship only: “Therefore, the purpose of this study was to extend the findings of Chua et al. (2021) by using a larger sample size to investigate the stressors that were experienced by the cohabiting couples, their psychological wellbeing during the lockdown, and to find out how the participants perceived the quality of their relationship with their cohabiting couple prior to and during the pandemic lockdown”

[4]There is information not provided in the methodology section.

1. Is this a secondary data analysis or first report of the study?

2. When and how was the survey conducted? How did you recruit participants (the sampling method). What was your data collection process?

3. Since you used the term "cohabiting" did you recruit samples in pairs from the same house?

4. How did you measure the before-lockdown and during lock-down status?

5. What statistical process did you use for your data analysis?

[5] Some wording issues. For example:

Pair Samples t Test should be Paired Samples t Test

Table 4. you repeated "Overall relationship quality before lockdown"

[6] Only paired t test was used in statistical data analysis.

Author Response

Dear Reviewer 1, we are grateful for your consideration of this manuscript, and we also very much appreciate your suggestions, which have been very helpful in improving the manuscript. All the comments we received on this manuscript have been taken into account in improving the quality.

Reviewer 2 Report

The COVID-19 pandemic has affected all aspects of people's lives in different countries, including marital relationships.

I agree with the authors that it is important to study relationships among cohabiting couples and compare them with similar studies in other countries. This allows researchers to identify common and culturally specific aspects of these relationships.

In the first part of the study, the authors analyzed the responses of East Malaysian cohabiting couples to an open-ended question and identified 5 main common stressors amid COVID-19 lockdown. For this part of the study, I have practically no issues, except that the open-ended question was addressed to an individual: “What stressors are YOU experiencing due to the СOVID-1 during the national lockdown?”. It would be possible to additionally ask about what stressors most influenced relationship in a couple.

In the second part of the study, the authors used questionnaires DASS-21 and PRQC Inventory. They then compare the results of these questionnaires before and during the lockdown. In this regard, the main question arises: did this study have 2 waves - before and during the lockdown?

If this is the case, then such a comparison is correct, but it is necessary to indicate the timing of the first and second waves of the study (this information is not in the article).

If there were no two waves in the study, and the respondents answered about their psychological states BEFORE the lockdown AFTER its beginning, then I do not consider this valid for a scientific research. In this case, I recommend that the authors refuse to compare the answers before and during the lockdown, and compare the results of these questionnaires in different subgroups: for example, men and women; spouses with different lengths of marriage, couples with different numbers or no children, etc.

The number of adjustments that I propose to make to the article depends on the answer to this question.

Author Response

Dear Reviewer 2, we are grateful for your consideration of this manuscript, and we also very much appreciate your suggestions, which have been very helpful in improving the manuscript. All the comments we received on this manuscript have been taken into account in improving the quality.
